# Manganese Modulates Metabolic Activity and Redox Homeostasis in Translationally Blocked *Lactococcus cremoris*, Impacting Metabolic Persistence, Cell Culturability, and Flavor Formation

Avis Dwi Wahyu Nugroho,[a,b,c] Berdien van Olst,[a,c,d] Stephanie Agnes Bachtiar,[a,b,c] Sjef Boeren,[a,d] Michiel Kleerebezem,[a,c]
Herwig Bachmann[a,b,e]

[a]TiFN, Wageningen, the Netherlands
[b]Microbiology Department, NIZO, Ede, the Netherlands
[c]Host-Microbe Interactomics Group, Wageningen University & Research, Wageningen, the Netherlands
[d]Laboratory of Biochemistry, Wageningen University & Research, the Netherlands
[e]Systems Biology Lab, Vrije Universiteit Amsterdam, Amsterdam, the Netherlands

**ABSTRACT** Manganese (Mn) is an essential trace element that is supplemented in microbial media with varying benefits across species and growth conditions. We found that growth of *Lactococcus cremoris* was unaffected by manganese omission from the growth medium. The main proteome adaptation to manganese omission involved increased manganese transporter production (up to 2,000-fold), while the remaining 10 significant proteome changes were between 1.4- and 4-fold. Further investigation in translationally blocked (TB), nongrowing cells showed that Mn supplementation (20 $\mu$M) led to approximately 1.5 X faster acidification compared with Mn-free conditions. However, this faster acidification stagnated within 24 h, likely due to draining of intracellular NADH that coincides with substantial loss of culturability. Conversely, without manganese, nongrowing cells persisted to acidify for weeks, albeit at a reduced rate, but maintaining redox balance and culturability. Strikingly, despite being unculturable, $\alpha$-keto acid-derived aldehydes continued to accumulate in cells incubated in the presence of manganese, whereas without manganese cells predominantly formed the corresponding alcohols. This is most likely reflecting NADH availability for the alcohol dehydrogenase-catalyzed conversion. Overall, manganese influences the lactococcal acidification rate, and flavor formation capacity in a redox dependent manner. These are important industrial traits especially during cheese ripening, where cells are in a non-growing, often unculturable state.

**IMPORTANCE** In nature as well as in various biotechnology applications, microorganisms are often in a nongrowing state and their metabolic persistence determines cell survival and functionality. Industrial examples are dairy fermentations where bacteria remain active during the ripening phases that can take up to months and even years. Here we investigated environmental factors that can influence lactococcal metabolic persistence throughout such prolonged periods. We found that in the absence of manganese, acidification of nongrowing cells remained active for weeks while in the presence of manganese it stopped within 1 day. The latter coincided with the accumulation of amino acid derived volatile metabolites. Based on metabolic conversions, proteome analysis, and a reporter assay, we demonstrated that the manganese elicited effects were NADH dependent. Overall the results show the effect of environmental modulation on prolonged cell-based catalysis, which is highly relevant to nongrowing cells in nature and biotechnological applications.

**KEYWORDS** *Lactococcus*, cellular redox status, fermentation, manganese, nongrowing

Address correspondence to Herwig Bachmann, herwig.bachmann@nizo.com.

The authors declare a conflict of interest. The project is organized by and executed under the auspices of TiFN, a public - private partnership on precompetitive research in food and nutrition. H.B. is employed by NIZO Food Research. The authors have declared that no competing interests exist in the writing of this publication. Funding for this research was obtained from Friesland Campina (Wageningen, The Netherlands), CSK Food Enrichment (Wageningen, The Netherlands) and the Top-sector Agri&Food.

**G**rowth and survival of microorganisms heavily relies on the environmental availability of metal cofactors, particularly for essential alkaline earth and transition metals such as magnesium, calcium, manganese, iron, cobalt, copper, and zinc. In this group, manganese is especially important because of its relatively high solubility, abundance, and distinctive redox abilities (1). In comparison with other biologically important redox-active metals, i.e., $Fe^{2+}$, $Mn^{2+}$ is a weaker electron donor or reducing agent (1). Consequently, cells can accumulate and tolerate high cytoplasmic concentration of free $Mn^{2+}$ (2) without negative redox outcomes under conditions that will normally promote formation of toxic free radicals through Fenton-type reactions (3). Based on structural similarity among other transition metals, only manganese is able to replace magnesium in its cofactor binding site and activate the corresponding enzymes which are ubiquitous in carbon, nucleic acid, and protein metabolism (1, 4, 5).

Therefore, intracellular manganese homeostasis is essential for optimal cellular activities. In bacteria, the intracellular $Mn^{2+}$ concentration is typically maintained relative to other metals as an inverse of the Irving-Williams series ($Mg^{2+}$ and $Ca^{2+}$ [weakest cofactor binding] $< Mn^{2+} < Fe^{2+} < Co^{2+} < Ni^{2+} < Cu^{2+} > Zn^{2+}$) (6, 7). This universal order predicts the stabilities of (transitional) metal complexes independent of the ligands (7), and highly influences the metal competition to cofactor binding sites that depends on both its abundance and affinity. To ensure specific metal cofactors are inserted to metalloenzymes, a cell finely tunes its intracellular metal pools by employing cytosolic metal sensors and transporters. In the case of manganese, two main transporters are reported in lactic acid bacteria (LAB), which are an ABC transport system (*mtsCBA*) and Nramp transporters (*mntH*) (8). Typically, intracellular manganese is maintained at micromolar levels, which is 1,000- to 10,000-fold lower than intracellular Mg2+ but 10,000- to 100,000-fold higher than other metals in the Irving-Williams series, with an exception for iron which is commonly present in comparable amount to manganese (6). The maintenance of intracellular manganese levels is especially relevant for cellular bioenergetics where various enzymes related to carbon metabolism, e.g., lactate dehydrogenase, phosphoglycerate mutase and fructose-1,6-bisphosphate phosphatase are either strictly Mn-dependent or highly stimulated by manganese (1, 9). Many bacterial superoxide dismutases which act as scavenger of reactive oxygen species also incorporates manganese in their active site (10). Therefore, manganese is generally considered to be crucial not only in survival under oxidative stress conditions, but also in ATP generation (10, 11).

In lactic acid bacteria, manganese supplementation has frequently been shown to contribute to cell growth and functionality during fermentation applications. The bioavailability of manganese has been found to enhance *in vitro* formation of flavors such as benzaldehyde (12) and the aldehydes derived from $\alpha$-keto acids, e.g., 3-methylbutanal (13). In the latter case, the conversion of branched-chain $\alpha$-keto acids (BCAAs) was reported to serve as a redox sink and the utilization of BCAAs can result in a marked increase of biomass, possibly due to additional ATP formation (14). On the other hand, lactococci grown without manganese supplementation have shown higher survival following a heat shock (15). Although the underlying mechanism to this observation is not known, it is plausible that manganese deprivation leads to stress responses that provide cross-protective resistance (16). Nonetheless, manganese supplementation is generally favored for microbial cultivation media despite these variations in physiological consequences in various species and growth conditions (17). While various studies have investigated the effect of manganese on growth and stress resistance, no studies investigated its effect on the metabolism of non-growing cells, which have distinct metabolic strategies and requirements. A nongrowing state is commonly encountered in various biotechnological applications such as in the production of pharmaceuticals (18, 19), fermented foods (20), or biofuels (21). It is especially relevant in various long-term fermentation processes such as cheese ripening where a significant portion of volatile flavor metabolites are generated for up to years after cell growth has ceased.

In the present study, we investigated the physiological and molecular (proteome) adaptation of *Lactococcus cremoris* to the presence and absence of manganese supplementation to a chemically defined medium. Furthermore, we investigated the role of

manganese in cellular survival and metabolic activity in growing and translationally blocked (TB) cells. The results indicate that cells in the absence of manganese can maintain their growth rate with relatively modest adjustment in cytoplasmic proteins compared with membrane transporters. However, in nongrowing, TB cells manganese omission led to a striking prolongation of acidification capacity, cell survival as well as maintenance of redox homeostasis. These observations demonstrate that manganese omission strongly influences the *L. cremoris* metabolism under TB conditions, while it does not appear to have apparent consequences for growth or physiology of *L. cremoris* during cultivation.

## RESULTS

**Manganese omission did not lead to changes in growth characteristics or metabolic end products.** Growth rate changes in many organisms are accompanied by metabolic shifts and they are suggested to reflect re-allocations in cellular protein investment and constraints on microbial growth (22, 23). In *Lactococci*, such metabolic shift has been described, where homofermentative acidification predominated by lactic acid occurs at a high growth rate but switches to heterofermentative acidification at a low growth rate (24). To investigate the requirement of manganese on growth, we removed manganese from the preparation of our standard chemically defined medium for lactococci. Remarkably, changes in the growth of strain NCDO712 was not detected in the absence of manganese. Because a carry-over effect from the previous growth medium might be sufficient to compensate for the lack of manganese, we further cultivated NCDO712 for four subcultures and a total of 25 generations to ensure complete removal of manganese. At the end of this subculturing, no apparent effect on the growth rates of NCDO712 was seen in relation to manganese availability (Fig. 1A). The growth rate remained high in the absence of manganese throughout the transfers and under excess or limited supply of lactose (Fig. S1), which implies that cells can grow with minute amounts of manganese available in their environment. In line with the high growth rate, the composition of produced organic acids remained unchanged and was predominated by lactic acid when manganese was omitted (Fig. 1B). These results suggest that Mn omission leads to no changes in flux (25) through the central energy generation pathway, i.e., glycolysis coupled to pyruvate conversion to lactic acid by lactate dehydrogenase (LDH), nor in the energy and redox state of the cells, i.e., ADP/ATP ratio, or NAD+/NADH ratio (26).

**Proteome adjustments to manganese omission mainly upregulated Mn transporters.** Growth data suggests that cells are apparently unaffected by manganese deprivation. To characterize the cellular responses to manganese omission from the growth medium, we performed a global proteome analysis. Manganese omission led to only 17 significantly differentially expressed proteins, encompassing 11 upregulated and six downregulated expression levels (Table 1). By far the most prominent proteome adaptation occurred in the expression of membrane transporters such as MntH, MtsA, and MtsB. There was an up to 2,000-fold increase of expression was observed for the ABC transporter MtsA upon the omission of manganese. The expression of proton dependent NRAMP-related manganese transporter MntH (plasmid encoded) increased under manganese omission by 4-fold. Next to these transporters, the putative manganese transporter llmg_1024 and llmg_1025 increased by 4-fold. This is comparable to the fold change observed for MntH and it may potentially be involved in the regulation of intracellular Mn concentration. Additionally, the genome-encoded MntH (*llmg*_1490) was not captured by differential analysis (Table 1) due to its level being below the detection limit with Mn supplementation. When taking into account the estimated minimum detection level in our proteomics data (4.8 LFQ intensity), this genome-encoded MntH increased by at least 20-fold upon Mn omission to an average LFQ intensity of 6.1, which is comparable with the upregulation of plasmid-encoded MntH$_P$ (Table S1). The upregulation of these Mn transporters indicates that cells detected Mn shortage despite no apparent changes in growth rate or acidification profile. However, it cannot be excluded that trace amounts of Mn that might be present as contaminants of medium constituents are compensating or masking any effects of Mn omission on growth. Moreover, it is

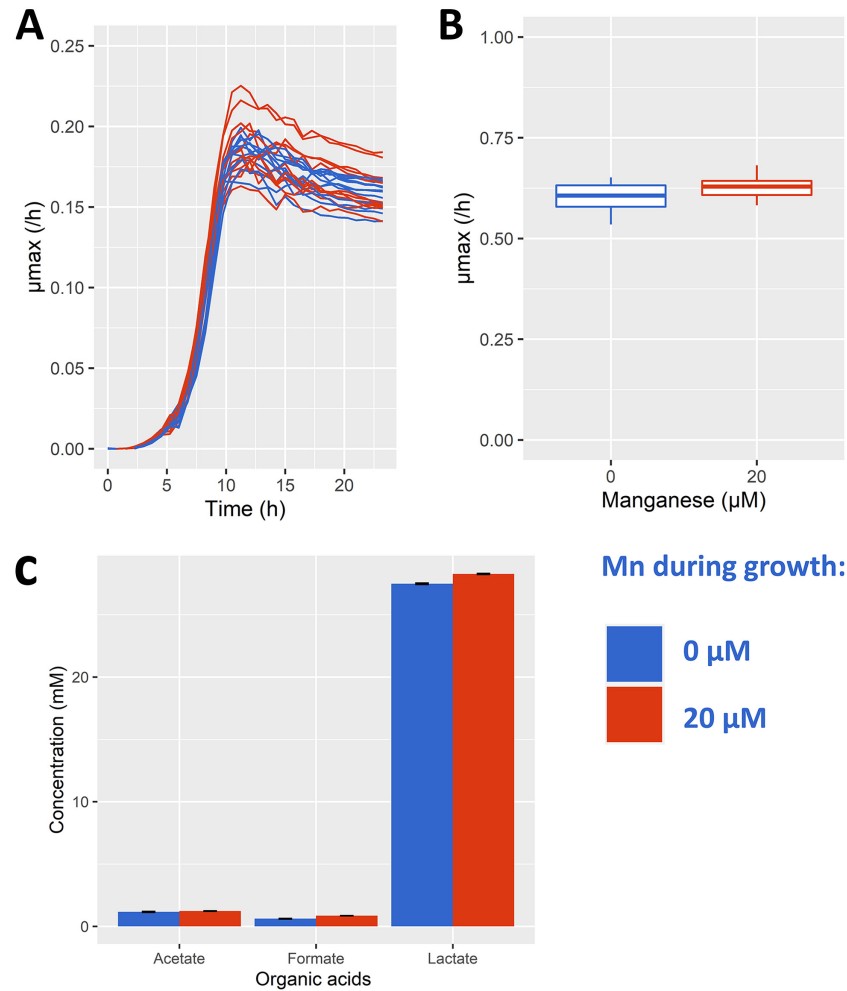

**FIG 1** *Lactococcus cremoris* NCDO712 was serially propagated four times (25 generations) in defined medium supplemented with lactose at excess (12.5 mM – growth stops due to acid accumulation) concentration in the presence (red) and absence (blue) of manganese (20 $\mu$M). The growth curve (panel A, $n = 8$), maximum specific growth rate (panel B, $n = 3$) and concentrations of organic acids (panel C, $n = 2$) are shown. Error bars indicate the standard deviation from the stated $n$ biological replicates.

unknown whether the upregulated transport functions would allow the cells to accumulate these trace amount to a sufficient intracellular level for growth.

Aside from these transporters, manganese omission also led to 12 more modest, but significant changes in cytoplasmic protein levels (ranging from 1.4- and 4-fold changes). Notably, the majority of these proteins are associated with redox metabolism and many catalyze NAD-dependent reactions. Manganese omission from the growth medium increased the expression of 2-dehydropantoate 2-reductase (3-fold) and a putative pyridoxamine 5′-phosphate oxidase (2.5-fold), while the expression of NADH oxidase (0.27-fold), aldehyde-alcohol dehydrogenase (0.41-fold), and a putative ferredoxin protein (0.48-fold) decreased. Moreover, manganese availability seems to correspond to changes in a few proteins related to stress response. For example, expression of the peptide methionine sulfoxide reductase (PMSR), which catalyzes the reduction of methionine sulfoxide in proteins back to methionine. This enzyme may protect cells against oxidative stress (27). It was found to be approximately 2-fold decreased in the absence of manganese. Conversely, manganese omission led to more than 3-fold increased expression of universal stress protein UspA, which is associated with resistance against various stresses. Intriguingly, aside from PMSR, differential expression of superoxide dismutase (MnSOD) and/or other oxidative stress related functions (Table S1) were not observed in the absence of manganese, illustrating a

**TABLE 1** Significant differentially expressed proteins in manganese omitted compared with manganese supplemented cultures[a]

| Protein names | Gene names | LFQ (+Mn) | LFQ (−Mn) | Fold change | -Log 10 of P-value |
|---|---|---|---|---|---|
| Manganese ABC transporter substrate binding protein | mtsA llmg_1138 | 4.9 ± 0.04 | 8.26 ± 0.11 | 2269.43 | 8.29 |
| Manganese ABC transporter ATP binding protein | mtsB llmg_1136 | 5.72 ± 0.22 | 7.77 ± 0.1 | 110.67 | 5.17 |
| Mn2+/Fe2+ transporter, NRAMP family | mntH pNZ712_01 | 6.43 ± 0.13 | 7.05 ± 0.04 | 4.19 | 3.69 |
| Uncharacterized protein | llmg_1025 | 6.5 ± 0.1 | 7.12 ± 0.04 | 4.15 | 4.14 |
| Putative membrane protein | llmg_1024 | 6.4 ± 0.09 | 6.97 ± 0.03 | 3.78 | 4.33 |
| Universal stress protein UspA | UspA | 7 ± 0.05 | 7.5 ± 0.06 | 3.20 | 4.54 |
| 2-dehydropantoate 2-reductase (EC 1.1.1.169) (Ketopantoate reductase) | panE llmg_1131 | 5.52 ± 0.15 | 5.99 ± 0.08 | 2.99 | 2.49 |
| Uncharacterized protein | llmg_2395 | 5.7 ± 0.1 | 6.11 ± 0.07 | 2.55 | 2.92 |
| Ribonuclease J (RNase J) (EC 3.1.-.-) | rnj llmg_0876 | 7.2 ± 0.01 | 7.51 ± 0.07 | 2.02 | 3.63 |
| Lipoprotein | plpB llmg_0336 | 6.82 ± 0.03 | 7.06 ± 0.05 | 1.75 | 3.30 |
| Ribonuclease J (RNase J) (EC 3.1.-.-) | rnj llmg_0302 | 7.32 ± 0.02 | 7.47 ± 0.03 | 1.40 | 3.20 |
| NADH oxidase (EC 1.6.-.-) | noxC llmg_1770 | 6.76 ± 0.14 | 6.19 ± 0.11 | 0.27 | 2.77 |
| Aldehyde-alcohol dehydrogenase | adhE llmg_2432 | 8.1 ± 0.05 | 7.72 ± 0.13 | 0.41 | 2.53 |
| Putative electron transport protein | llmg_1916 | 6.9 ± 0.1 | 6.58 ± 0.04 | 0.48 | 2.64 |
| Peptide methionine sulfoxide reductase MsrA (Protein-methionine-S-oxide reductase) (EC 1.8.4.11) | pmsR msrA llmg_2281 | 6.25 ± 0.06 | 5.97 ± 0.07 | 0.52 | 2.71 |
| Glycine betaine/proline ABC transporter (EC 3.6.3.32) | busAA llmg_1048 | 8.17 ± 0.04 | 7.92 ± 0.02 | 0.56 | 4.22 |
| Glycine betaine-binding protein | busAB llmg_1049 | 7.7 ± 0.03 | 7.46 ± 0.02 | 0.59 | 4.62 |

[a]Proteins were selected based on cutoff parameters of s0 = 0.01 and a false discovery rate (FDR) of 0.05. Label free quantitation (LFQ) values represent the average from four biological replicates.

lack of prominent changes in the oxidative stress levels experienced by these cells. Overall, the changes in proteome data are dominated by major changes in Mn-transport proteins, and more modest changes in a number of cytosolic proteins that suggest that cells might experience a shift in the redox balance.

**Acidification is maintained for a prolonged period in TB cells only in the absence of manganese.** The proteome data implied that cells cultured without Mn addition might be in an altered physiological state, which may depend on the intracellular Mn levels and/or $NAD^+$/NADH availability to maintain metabolic fluxes. We investigated if cellular adaptation to manganese omission affects central metabolism, which we performed in a previously described model system of non-growing cells with inhibited protein synthesis (28). In this system we can follow acid production of nongrowing cells with continuous measurements for up to 3 weeks, compared with growing cells which fully acidifies in a few hours. Inhibition of protein synthesis with the antibiotic erythromycin enables the preservation of protein levels, e.g., manganese transporters. It allows comparing of cells grown in the presence or absence of manganese, while manganese concentration in the assay medium can be precisely adjusted. In line with the growth data, lactic acid was still the predominant fermentation end product, making up more than 90% (Cmol-based) of the produced organic acids (Fig. S2A) during prolonged incubation irrespective of the treatments. The calculated lactic acid production from the continuous pH measurement (Fig. 2B) is in good agreement with the HPAEC measurements (Fig. 2A).

In the presence of manganese, acidification by TB cells was initially approximately 1.5-fold faster (Fig. S2B) but stagnated within 24 h (Fig. 2A and 2B, left panel) reaching a relatively low final lactate concentration (~ 4 mM), and a final pH of 6.16. In contrast, when cells were transferred to assay medium containing no manganese, acidification continued at a lower rate for more than a week, reaching drastically higher final lactic acid concentrations (~ 25 mM) and a lower final pH of ~ 4.0. The latter conditions (25 mM lactic acid and pH 4.0) are likely the environmental conditions that prevented further acidification (Fig. 2A and B). Moreover, in the absence of Mn in TB-cell assay (Fig. 2B), Mn-precultured cells produced 20 mM lactic acid in 100 h compared with 150 h when cells were precultured without Mn, potentially as a result of trace amount of Mn carry over. Importantly, these results demonstrate that the absence of manganese in these acidification assays prominently changes the persistence of flux through the central energy-generating pathway. This seems to be irrespective of the accompanying

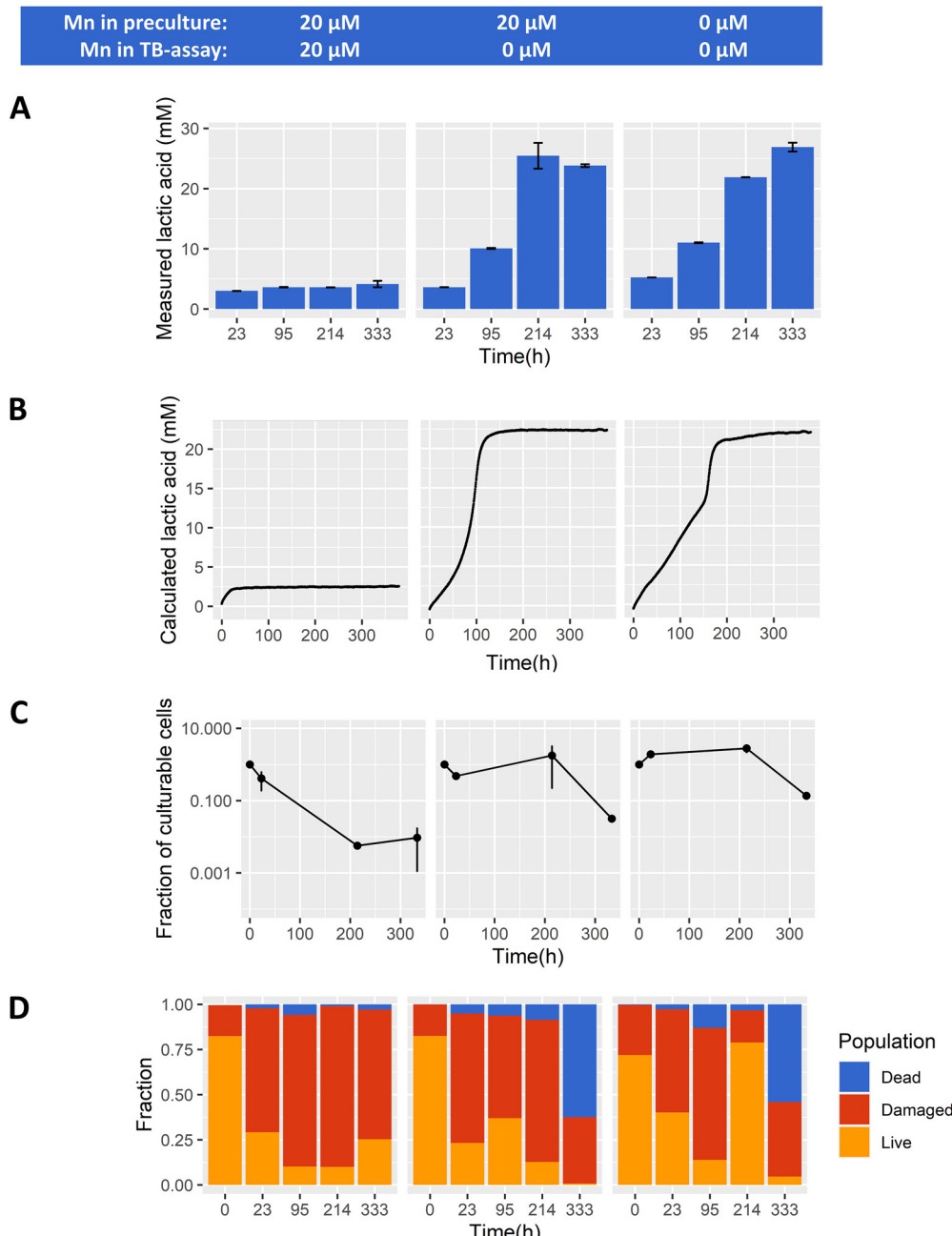

**FIG 2** *Lactococcus cremoris* NCDO712 (*n* = 3 biological replicates) was precultured in the presence (left and middle panel) and absence (right panel) of manganese (20 $\mu$M). Cells (2.5E + 07 cells/mL) were transferred into fresh medium containing erythromycin (5 $\mu$g/mL) and 20 $\mu$M manganese (left panel) or 0 $\mu$M manganese (middle and right panel). Concentration of lactic acid (panel A) was measured with HPAEC at selected time points. Continuous measurement of medium pH to calculate lactic acid production overtime can be seen in panel B (replicates behaved nearly identically). Average population fractions based on membrane integrity (panel C) was measured for dead (blue), damaged (red), and live (orange) cells throughout incubation. Fractions of culturable cells based on plate counts can be seen in panel D. Error bars indicate the standard deviation.

cellular proteome changes as demonstrated by preculture both in the presence or absence of Mn showing prolonged activity following transition to Mn-omitted nongrowing assay medium (Fig. 2A and B middle and right panel). Despite of the prominent role of manganese in oxidative stress tolerance in many organisms, there were no indications that this stagnated acidification was explained by substantial difference in oxidative stress levels in the absence or presence of Mn in this TB assay (Fig. S3). Overall

this data shows that the omission of manganese during sugar conversion of translationally blocked cells allows for a much longer persistence of acidification and therefore a higher total product formation.

**Manganese induces the appearance of viable but nonculturable populations in TB cells after acidification stagnates.** The observed stagnation of acidification within 24 h for Mn supplemented conditions implies that the generation of ATP through glycolysis would also stagnate, which could also affect the viability and integrity of the cells. In this context it is relevant to note that *Lactococci* is known to remain metabolically-active for prolonged periods of carbon starvation, e.g., more than 3.5 years, in a viable but nonculturable (VBNC) state (29). To investigate the influence of manganese on cellular integrity and culturability during acidification in TB cells, we determined the CFU and membrane integrity over time. Within 9 days of prolonged incubation, nongrowing cells incubated with manganese showed an approximate 100-fold reduction in culturable cells (Fig. 2C). This is sharply contrasted by results obtained for cells incubated without manganese, where culturability was maintained close to 100% in the same time frame. Under the conditions used, rapidly declining culturability of the cells incubated in the absence of manganese was only observed after 2 weeks of incubation, which is likely the consequence of the combined stress of low pH and increased lactic acid concentrations. These results are in good agreement with our previous observations that an approximate 40-fold decline of viability was observed under similar conditions after approximately 2 weeks (28). Analogous to the acidification observations presented above, the presence or absence of manganese during the preculturing and the corresponding proteome adaptations did not significantly influence the culturability results we obtained.

Membrane integrity analysis (Fig. 2D) of the same time series revealed a prominent decline of the subpopulation with an apparent intact membrane ("live") that increasingly progressed toward the subpopulation characterized by slightly damaged ("damaged") or severely damaged ("dead") membrane integrity over the course of incubation. Under all conditions, the damaged subpopulation became more predominant over time and on average ranged from 60% to 80% of the total population in cells incubated between 23 h and 214 h. Notably, in the absence of manganese the subpopulation classified as "dead" increased to approximately 60% of the total population after 333 h of incubation, whereas less than 5% of the total population was classified as "dead" when incubated for the same time in media containing manganese. Analogous to the culturability results, this drastic decline of viability is likely due to prolonged exposure to low pH and high lactic acid concentrations inducing increasing cell damage that coincides with loss of culturability (and viability). Conversely, the observed decline in culturability of cells incubated in the presence of Mn was very poorly reflected by an increasing subpopulation of cells characterized as "dead" according to these membrane-staining procedures. Apparently, the loss of culturability of these cells is not related to loss of cellular integrity but could be related to an unbalanced metabolism because of their high rate of acidification. Potential metabolic consequences of rapid acidification could induce stagnation of acidification and loss of culturability. This could be related to an excessive increase of the ATP/ADP balance, where the depletion of the intracellular ADP pool halts glycolytic flux. Alternatively, the disruption of the intracellular redox balance (NADH/NAD+ ratio) could lead to depletion of either form of this cofactor that would also effectively halt glycolytic flux and/or lactate formation. Loss of either ATP/ADP or NADH/NAD+ homeostasis may also negatively affect culturability by creating an inability to re-initiate energy-generation or biosynthesis pathways required for regrowth.

**Manganese leads to NADH depletion in acidifying TB cells.** To investigate whether manganese induces the proposed ADP depletion and thereby stagnates acidification in TB cells, we used strain MG1363 (a prophage-cured and plasmid-free derivative of strain NCDO712) harboring pCPC75::atpAGD (30). This strain constitutively overexpresses the F(1) domain of the membrane-bound F(1)F(0)-ATPase, which modulates intracellular energy levels by accelerating the ATP to ADP conversion (30). However, in the presence of manganese TB cells of this F(1)-ATPase overexpressing strain displayed the same stagnation of

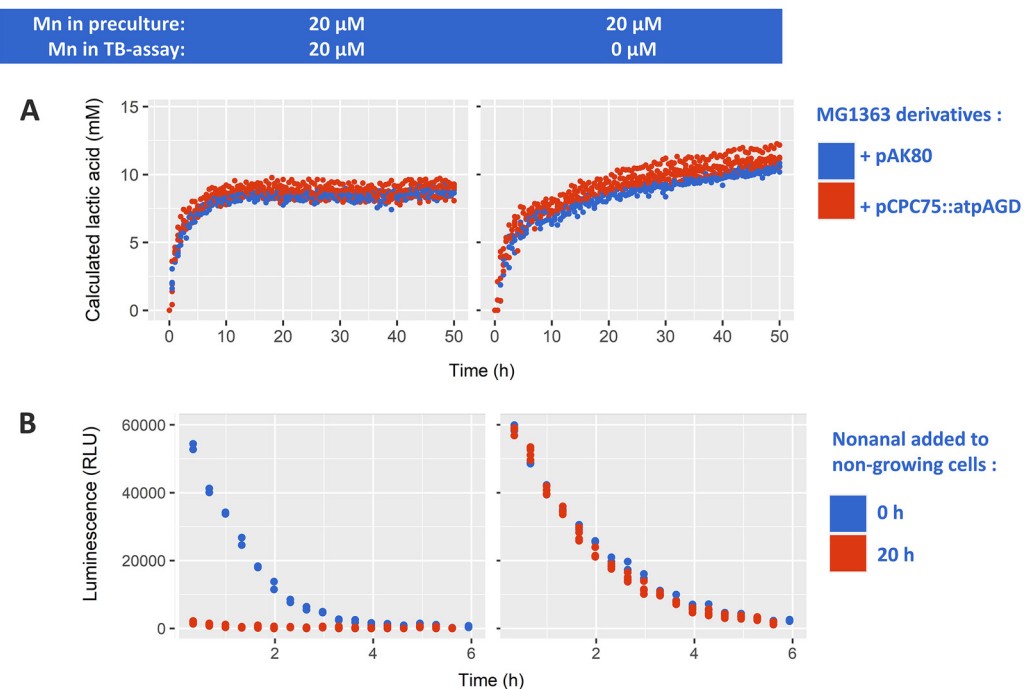

**FIG 3** Cells precultured with 20 μM Mn were transferred into TB assay with 20 μM Mn (left column) or 0 μM Mn (right column). (A) *Lactococcus cremoris* MG1363 that harbors empty vector pAK80 (upper) or F1-ATPase encoding pCPC75::atpAGD (lower) at 2.0E + 07 cells/mL were analyzed for continuous measurement of medium pH to calculate lactic acid production overtime. (B) *Lactococcus cremoris* MG1363 (pNZ5519) encoding bacterial *luxAB* luciferase was analyzed for luminescence signal maintenance when starting the reaction after 0 h (blue) and 20 h (red) of incubation with TB. Cells were incubated at 1.0E + 07 cells/mL and concentrated to 1E + 08 cells/mL prior to luminescence detection. Experiment was carried out at least with three biological replicates.

acidification within 24 h as the control strain (MG1363 harboring the empty vector pAK80), whereas manganese omission allowed continued acidification at a reduced rate for weeks (Fig. 3A). These results demonstrated that preventing the postulated ATP accumulation by its conversion to ADP through the F(1)-ATPase failed to sustain prolonged acidification. This indicates that the disruption of the ATP/ADP homeostasis is not the mechanism underlying the observed phenomenon.

To investigate whether a redox balance disruption is able to explain the manganese-induced stagnation of acidification, we used strain MG1363 harboring pNZ5519 (31) which constitutively expresses *luxAB*, a bacterial luciferase of *Vibrio harveyi*. This luciferase catalyzes the reaction of a long chain aldehyde, e.g., nonanal, oxygen, and reduced flavin mononucleotide ($FMNH_2$) to form a carboxylic acid, FMN, and light (490 nm). Regeneration of intracellular $FMNH_2$ is dependent on the availability of NADH, which is thereby required to maintain the luciferase reaction and the corresponding detection of the luminescence signal. In an initial experiment, the LuxAB substrate nonanal was provided immediately after transferring cells to the TB acidification conditions. In this experiment, the initial luminescence signal was approximately equal in absence and presence of manganese, and subsequently declined over time, to become undetectable after approximately 6 h (Fig. 3B). This result indicates that the high NADH demand of the luminescence reaction effectively drains the cellular NADH pool, which leads to the rapid decline of the luminescence signal over time. Notably, the decline rate of the luminescence signal was higher when manganese was supplemented and reached undetectable luminescence levels 2 h earlier compared to the condition when manganese was omitted, suggesting that NADH is depleted more rapidly when manganese is present. In a follow-up experiment, TB-cell suspensions were left to acidify for 20 h prior to the addition of nonanal to initiate the luminescence reaction (Fig. 3B). Under these conditions, the impact of manganese presence in the incubation medium was very pronounced, where the condition lacking manganese generated an initial luminescence level and a subsequent signal-decline

curve that were very similar to those observed when nonanal was added from the start, whereas in the presence of Mn there was hardly any detectable luminescence signal (Fig. 3B). These results suggest that NADH is depleting during acidification in the presence of Mn, whereas intracellular levels of NADH and redox homeostasis are maintained in the absence of manganese. Taken together these experiments show that manganese-mediated acceleration of acidification correlates with a disruption of redox homeostasis rather than energy homeostasis (ATP/ADP), leading to depletion of NADH and thereby stagnating acidification and possibly inducing a VBNC state. This loss of redox homeostasis is not seen in the absence of manganese where NADH pools are apparently kept constant and acidification can be sustained for weeks (28) in these TB-cell suspensions.

**NADH-dependent conversion of aldehydes to alcohols increases in TB cells upon manganese omission.** Next to acidification, NADH depletion might influence or re-route other (industrially relevant) metabolic pathways that are dependent on the redox state of the cell, such as branched-chain amino acid catabolism. This catabolic pathway is initiated by the transamination of the amino acid leading to the formation of the corresponding $\alpha$-keto-iso-caproic acid (KICA), which serves as the major metabolic precursor that can be converted to three different intermediate products. Only the keto acid conversion to the corresponding aldehyde intermediate, e.g., 3- or 2-methylbutanal, is independent of NAD+ or NADH as a cofactor and is also enhanced by Mn (13). The other keto acid conversions include the NADH-dependent conversion to $\alpha$-hydroxyisocaproic acid via hydroxyacid dehydrogenase and the NAD+ and CoA dependent conversion to isocaproyl-CoA via ketoacid dehydrogenase.

Volatile analysis from TB-cultures shows that the production of 3- and 2-methylbutanal within the first 23 h to 95 h was comparable throughout all conditions (Fig. 4). However, in the presence of Mn, 3- and 2-methylbutanal continued to accumulate during 2 weeks of incubation to reach an average peak area of 7.5E + 06 (arbitrary unit), which is approximately 3-fold higher than the maximum level reached in the absence of Mn. In contrast, BCAA-derived aldehydes were not further accumulating after the first 23 h in the absence of Mn, and actually declined after 95 h, suggesting their utilization in aldehyde-consuming reactions. A known reaction for aldehyde conversion leads to formation of the corresponding alcohol (3- and 2-methylbutanol) catalyzed by aldehyde-alcohol dehydrogenase, which is NADH dependent. We found that in the absence of manganese, the maximum peak area of 3-methylbutanol and 2-methylbutanol were increased approximately 10-fold and 5-fold, respectively, in the conditions without manganese compared with those where manganese was supplemented. In the presence of Mn, it is apparent that the reaction cascade stalls at the aldehyde formation, and fails to convert to the alcohol, which agrees with the proposed NADH depletion.

## DISCUSSION

In this study, we evaluated how manganese influenced the physiology of *L. cremoris*, and show that manganese omission from the growth medium did not impose a measurable growth rate reduction, while bringing a substantial survival advantage upon translational blocking. We observed that the adaptation of the cellular proteome to growth conditions that lack manganese mainly involves the upregulation of Mn importers. This may allow for the accumulation of minute levels of Mn contaminations from medium constituents inside the cell to achieve Mn levels that support a high growth rate. Although we did not determine intracellular Mn concentrations, it has previously been reported that *Lactococcus cremoris* MG1363 accumulated up to 0.7 mM Mn intracellularly during growth in the medium that was also employed in this study (32). The study implied that manganese is required for lactococcal enzyme activities, making our finding that we can omit this metal from the medium without consequences in terms of growth or central energy metabolism even more striking. In addition, the results we report here are also contrasting the apparent dependency for Mn in other lactic acid bacteria species, to sustain rapid growth and oxygen tolerance (33, 34).

Cytoplasmic manganese is important in lactococci, which is supported by the extensive transport systems dedicated for Mn homeostasis. Four Mn transport systems were

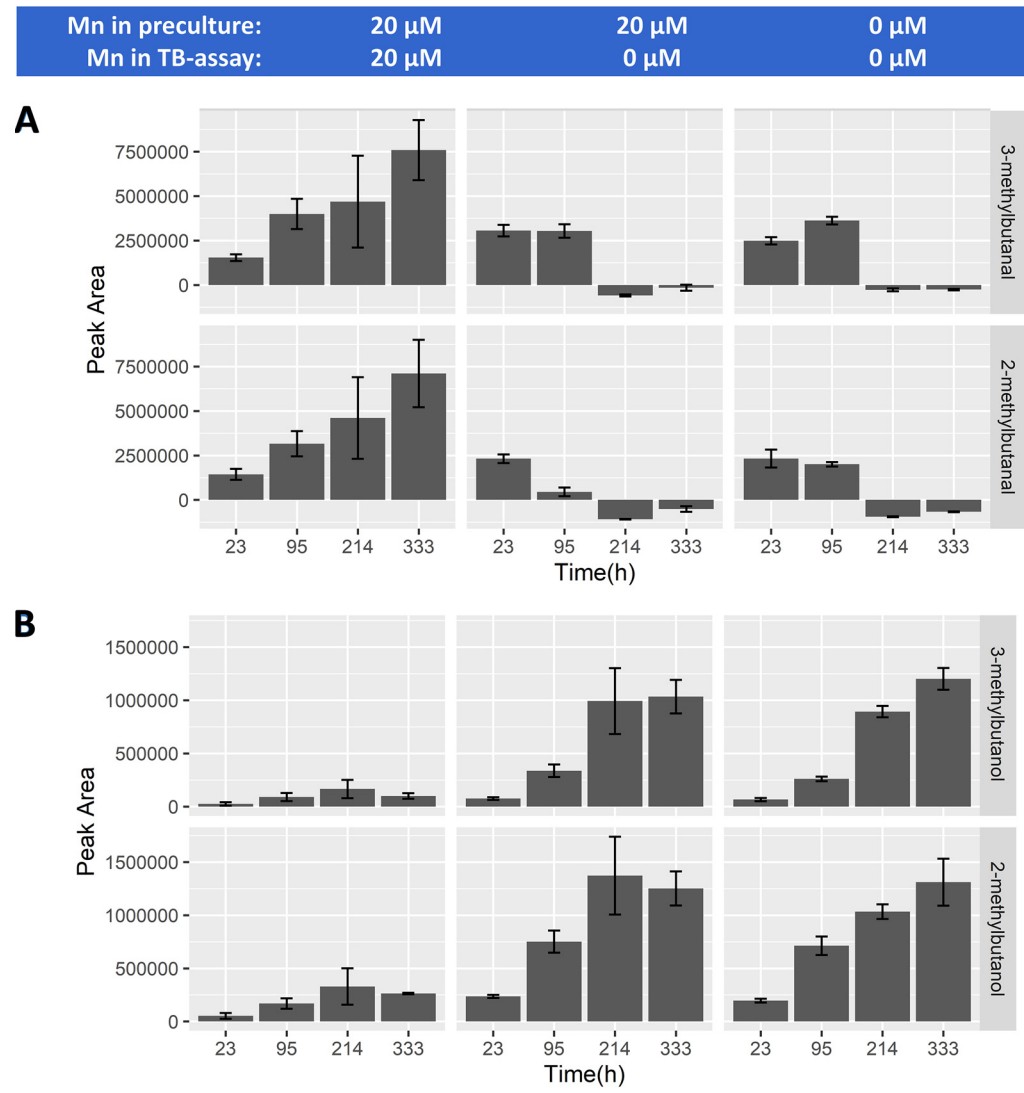

FIG 4 *Lactococcus cremoris* NCDO712 was precultured in the presence (left and middle panel) and absence (right panel) of manganese (20 μM). Cells (2.5E + 07 cells/mL) were transferred into fresh medium containing erythromycin (5 μg/mL) and 20 μM manganese (left panel) or 0 μM manganese (middle and right panel). GC-MS peak areas of 3-methylbutanal and 2-methylbutanal (panel A) as well as 3-methylbutanol and 2- methylbutanol (panel B) were measured throughout incubation. Error bars indicate the standard deviation from three biological replicates.

suggested from the MG1363 genome which includes the chromosomal NRAMP-family transporter (encoded by *mntH*), an ABC transporter (encoded by *mtsAB*), and the putative Mn-Fe importer (encoded by *llmg1024-1025*). Additionally, *L. cremoris* strain NCDO712 used in this study also harbors six plasmids, one of which encodes an extra copy of NRAMP-family transporter MntHp (encoded by *mntH$_p$*). Next to the characterized transport systems, the proteins encoded by *llmg_1024-1025* (domain PF01988; detected with E-value 3.1E-41 and 3.1E-40, respectively) show the presence of a VIT1 domain that is also found in various Fe2+/Mn2+ import systems. The upregulation we found for these genes support the role of this protein in Mn import. From an evolutionary perspective, dairy lactococci need to acquire manganese, of which the availability is limited in their natural environment. In bovine milk, Mn concentrations are reported to vary between 4 and 6.5 μM (35) which is in the same order of the concentration supplied in the present study. The total concentration of Mn in cheese was found at a similar level, which was between 5 and 20 μmole per kg cheese (36), but its bioavailability might be influenced by the pH, salt concentration, association with other (metal) ions or casein micelles, water activity, and various other

factors (37). The availability of redundant transporter systems are likely to ensure sufficient uptake under dynamic environmental conditions.

It was unexpected that in nongrowing cells, we found that manganese supplementation at a physiological concentration for growth led to rapidly stagnating acidification and a culturability decline. This effect on acidification may result from the manganese-mediated modulation of the activity of enzymes in the central energy generating pathway. Within this metabolic pathway, various enzymes were reported to be dependent or activated by manganese such as FBPase, PGM, and LDH (1, 9, 38). As a result, manganese addition might affect the homeostasis of this pathway. The balance of ADP/ATP and NAD+/NADH are especially important as both have been implicated in controlling metabolic flux as well as fermentation end product profiles from homolactic to mixed acids (26, 39). We demonstrated that in our setup the effects on central metabolism were associated with NAD+/NADH rather than ADP/ATP disbalance.

Manganese deficiency might not be fully compensated by overproduction of Mn importers as demonstrated by the changes of NADH-utilizing enzymes in the absence of Mn. If NADH is a rate-limiting factor, such changes might be an effort to tune the flux through those reactions and thereby preventing an excessive drain of the intracellular NADH pool. Alternatively, regulating the level of these enzymes at low intracellular NADH concentrations may be necessary to maintain their flux at the required level. In contrast, translational blocking disallows proteome changes that may serve as cell's coping mechanism against NAD imbalance. Without proteome adjustment, a slight disbalance in NAD(H) regeneration might lead to a substantial depletion of NAD(H) over a prolonged period of time, which potentially explains the stagnating acidification of nongrowing cells. Consequently, the lack of NADH also blocks other pathways that rely on its availability such as alcohol production in BCAA catabolism as well as $FMNH_2$ regeneration required for luminescence reaction (Fig. 5). Nevertheless, it is remarkable that the long term conversion of keto acids to aldehydes in this study was maintained even when central energy-generating pathway has halted for an extended period, implying that cells are likely to maintain a sufficiently high energetic state to import amino acids and that NADH depletion rather than ATP depletion corresponds to the emergence of VBNC state. In yeasts, it has been reported that NAD(P)H depletion is associated with VBNC state resulting from sulfite exposure (40, 41). In nongrowing cells of retentostats, where the carbon source is continuously supplied albeit rapidly utilized (42), the production of various aldehydes from amino acid degradation such as benzaldehyde, benzeneacetyladehyde, 3-(methylthio)-propanal was not only highly correlated with the low growth rate but also with the increased loss of culturability on agar. While intracellular NADH concentration was not measured in these retentostat studies, it could be that NADH disbalance coincides with the emergence of VBNC populations and coinciding accumulation of aldehyde-volatiles under those conditions.

Our study is relevant for biotechnological and fermentation purposes, especially when the metabolism of non-growing cells that rely on cofactor recycling is of interest. As demonstrated, the prolonged stability of acidification and NADH regeneration are potentially crucial for the transition toward nonculturable state. Such transition is potentially relevant for various applications where cells are stored under suboptimal conditions such as commercial starters and probiotic products. In line with other investigations of lactococci at near-zero growth (43), the distinct end-metabolite and its accumulated formation by viable but nonculturable lactococci substantiate their potential important role in flavor formation during ripening. This was exemplified by BCAA-derived volatiles that are particularly critical for the flavor characteristics of fermented foods. Aldehyde intermediates from BCAA catabolism are more potent flavor volatiles than their alcohol derivatives with an odor threshold that differs by two orders of magnitude (44). The prolonged aldehyde formation in cells that no longer acidify and presumably no longer generating substantial levels of ATP imply that this metabolite formation is uncoupled from cellular energy status and growth and potentially very suitable for cell factory applications. Next to volatile production, the present study

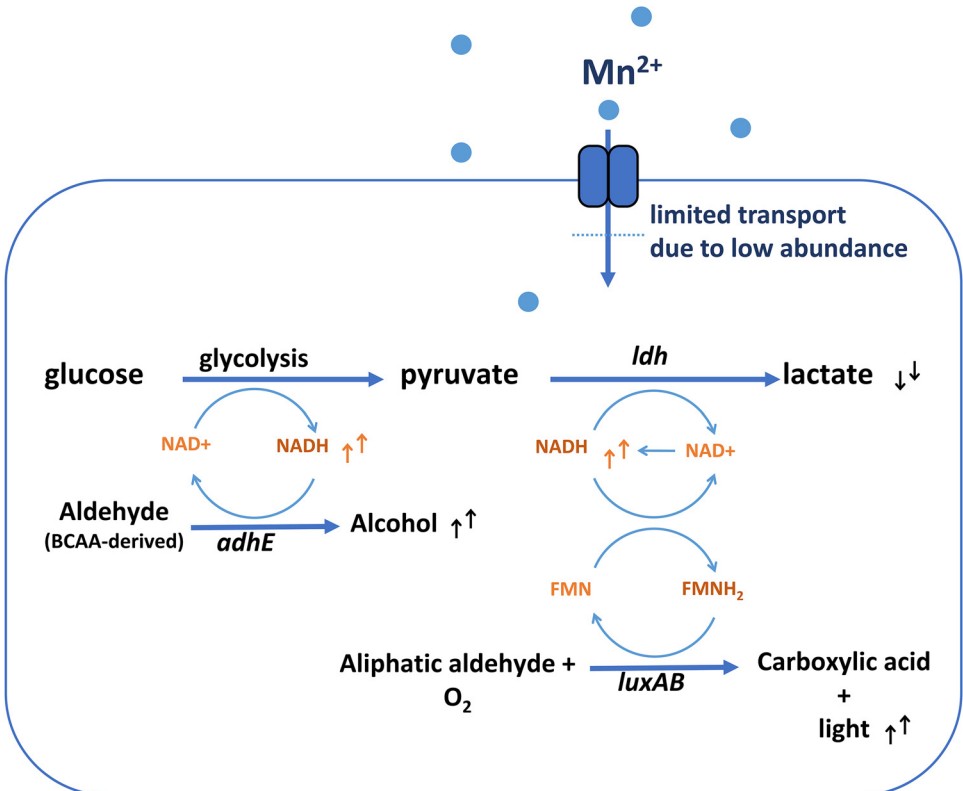

**FIG 5** Schematic simplification of the effect of manganese omission on the metabolism of TB-*L. cremoris* NCDO712. Limited manganese import might reduce glycolysis flux or LDH activity, but prevents NADH depletion, and allows other NADH-dependent reactions to take place for a prolonged period. Increase (↑) or decrease (↓) of metabolic compounds measured upon manganese omission are indicated.

further highlights the relevance of our high-throughput nongrowing model system for the investigation of starter cultures in cheese ripening where flavor formation by non-acidifying VBNC cells with limited protein synthesis can be mimicked. Finally, understanding the factors that influence the stability of prolonged metabolism through the presented TB model may provide new approaches in modulating the yield of desired compounds produced by nongrowing cells.

## MATERIALS AND METHODS

**Strain and Mn-omission cultivation.** *Lactococcus cremoris* NCDO712 (45), MG1363 (46), MG1363 (pAK80) (30), MG1363(pCPC75::atpAGD) (30), and MG1363 (pNZ5519) (this study, Supplementary Methods 1) were grown on chemically defined medium for prolonged cultivation (CDMPC, Supplementary Methods 2) (47) at 30°C without aeration. Strains were precultured in the presence or absence of Mn for 25 generations (four direct transfers with a 100-fold dilution) to minimize carry-over effects. Details on strain-specific ingredients and growth rate measurement and can be found in Supplementary Methods 3 and 4.

**Proteome analysis.** Proteome samples were harvested in quadruplicates from exponentially growing cultures of strain NCDO712 that was precultured for 20 generations in the presence or absence of manganese. Protein extraction and analysis were performed as described previously (48). Details and modifications to the proteomics methods can be found in Supplementary Methods 5. The mass spectrometry proteomics data have been deposited to the ProteomeXchange Consortium via the PRIDE (49) partner repository with the data set identifier PXD030123.

**Acidification and luminescence measurements of TB cells.** Long-term analysis of metabolite production was performed as described earlier (28). Exponentially growing cells pre-cultured in the presence or absence of Mn were harvested and resuspended at a density between 1E + 07 and 2.5E + 07 cells/mL in their corresponding growth medium supplemented with erythromycin (5 $\mu$g/mL) and 10 $\mu$M 5(6)-carboxyfluorescein (Sigma-Aldrich 21877), with or without Mn (20 $\mu$M). For volatile production measurement, aliquots of TB-cultures were transferred to sterile GC-MS vials. Time-course samples were analyzed for organic acids, volatiles, viability, and membrane integrity (Supplementary Methods 6 and 7).

Luminescence measurement was performed as described earlier (50). Aliquot of TB-*L. cremoris* MG1363 harboring pNZ5519 (1E + 07 cells/mL, precultured without manganese) was concentrated at selected time points of incubation by centrifugation and resuspension in less volume of supernatant to a concentration of

1E + 08 cells/mL. Nonanal 1% in silicon oil was supplied as reaction substrate and added either into empty wells or the empty space between the wells of the microplate. A regular lid was used to cover the microplate which allows sufficient oxygen exposure required for the luciferase reaction. Luminescence was determined at 20-min intervals over a period of 6 h after nonanal addition in a Genios microplate reader (Tecan, Zurich, Switzerland). The gain was set at 200 and integration time was set at 1,000 ms.

**Data availability.** The proteome data associated with this manuscript can be accessed in the Pride database (http://proteomecentral.proteomexchange.org/cgi/GetDataset) with the dataset identifier PXD030123.

## SUPPLEMENTAL MATERIAL

Supplemental material is available online only.

**SUPPLEMENTAL FILE 1**, XLSX file, 0.4 MB.

**SUPPLEMENTAL FILE 2**, PDF file, 0.5 MB.

## ACKNOWLEDGMENTS

We thank Roelie Holleman for the HPLC measurement of organic acids, Wilma Wesselink for the HS-SPME GC-MS measurement of volatiles, Peter Ruhdal Jensen who kindly provided strain MG1363(pAK80) and MG1363(pCPC75::atpAGD), as well as Jacques Vervoort for the constructive discussion.

The project is organized by and executed under the auspices of TiFN, a public–private partnership on precompetitive research in food and nutrition. H.B. is employed by NIZO Food Research. The authors have declared that no competing interests exist in the writing of this publication. Funding for this research was obtained from Friesland Campina (Wageningen, the Netherlands), CSK Food Enrichment (Wageningen, the Netherlands), and the Topsector Agri & Food.

A.D.W.N., M.K., and H.B. conceived and designed the study; A.D.W.N., B.v.O., S.B., and S.A.B. carried out the experiments; all authors analyzed the data; A.D.W.N., M.K., and H.B. wrote the paper.

This study was funded by the Top Institute Food & Nutrition (TIFN, Program 16MF01, Wageningen, the Netherlands).

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
