## [Reviewer comments · Microbiology Spectrum]

Microbiology Spectrum

Manganese modulates metabolic activity and redox homeostasis in translationally-blocked *Lactococcus cremoris*, impacting metabolic persistence, cell-culturability, and flavor formation.

Avis Nugroho, Berdien van Olst, Stephanie Bachtiar, Sjef Boeren, Michiel Kleerebezem, and Herwig Bachmann

Corresponding Author(s): Herwig Bachmann, NIZO Food Research, TIFN

Review Timeline:

Submission Date:	December 26, 2021
Editorial Decision:	January 20, 2022
Revision Received:	April 20, 2022
Accepted:	April 26, 2022

Editor: Eva Sonnenschein

Reviewer(s): The reviewers have opted to remain anonymous.

Transaction Report:

DOI: <https://doi.org/10.1128/spectrum.02708-21>

January 20, 2022

Dr. Herwig Bachmann
NIZO Food Research, TIFN
Kernhemseweg 2
Ede 6718ZB
Netherlands

Re: Spectrum02708-21 (Manganese modulates metabolic activity and redox homeostasis in translationally-blocked *Lactococcus cremoris*, impacting metabolic persistence, cell-culturability, and flavor formation.)

Dear Dr. Herwig Bachmann:

Link Not Available

Sincerely,

Eva Sonnenschein

Journals Department
Reviewer comments:

Reviewer #1 (Comments for the Author):

Manganese modulates metabolic activity and redox homeostasis in translationally-blocked *Lactococcus cremoris*, impacting metabolic persistence, cell-culturability, and flavor formation." (control no. Spectrum02708-21)

This work describes the impact of manganese limitation on the physiology of the model lactic acid bacterium *Lactococcus lactis*. Interestingly *L. lactis* can sustain strong growth in the absence of Mn. Proteomic analysis revealed Mn transporters to be upregulated upon Mn starvation, as well as several intracellular proteins involved in redox reactions. Using cells that are translationally blocked, it was found Mn omission leads to initially slower but more sustained lactic acid production and a greater conversion of aldehydes to alcohols during extended incubation. It was shown that the presence of Mn leads to NADH depletion which coincides with a significant loss of culturability.

Overall this manuscript provides new insights into the role Mn plays in cell physiology which significantly impacts *L. lactis*

acidification, flavour compound production and survival. The model system established likely mimics the *L. lactis* non-growing status in a cheese ripening situation, and therefore is of interest to food fermentation applications and holds potential for future modulation of flavour compounds (e.g. through Mn addition to milk used in cheese making). The manuscript is very well written and data is clearly presented, and I don't have any major concerns.

Specific comments:

- Line 255 and 264 - re-order the Figure 2C and 2D panels so that they are referred to in the text in the correct order.
- Some references are missing volume and page numbers (e.g. refs 9, 10, 40, 43, 44)

Reviewer #2 (Comments for the Author):

The authors describe the impact of the presence of manganese in the growth medium, or the omission of Mn, on growth, acidification and survival of *Lactococcus cremoris*. A proteomic experiment shows that ABC- and NRAMP-type uptake systems are strongly up-regulated in the absence of Mn, as would have been expected. Since lactate dehydrogenase (LDH) should be activated by Mn(II) (or Co(II)) as in *Streptococcus faecalis*, absence of Mn should lead to missing activation of LDH, decreased acidification of the growth medium and accumulation of NADH. Surprisingly, this was not observed. The cells grew not differently in the presence or absence of Mn within 25 h, and the concentration of produced lactate was similar. Homofermentative lactate fermentation was done, nearly no acetate or formate produced by alternative fermentation pathways. The paper shifts subsequently to "translation-blocked" cells that were inhibited by erythromycin. Now, the cells produced only low amounts of lactate in the presence of Mn, did not show high numbers of dead cells in a life/dead straining despite a 100-fold decrease in cfu, indicating a vbnc state. These cells had only low NADH concentrations after 20 h. In contrast, they continued to produce lactate over days in the absence of Mn, possessed sufficient NADH after 20 h and contained a large portion of dead cells accompanied by a decrease of cfu after many days.

1. The Methods section is very small, in the paper and the supplement. The exact composition of the growth medium should appear in the supplement. The Methods in the paper should define "TB-cells" already here; it is defined too late in the Results part.
2. It could be very helpful to use an ICP-MS to measure the metal content of the growth medium "with omitted Mn" and also in the cells! NRAMP systems are of a broad substrate specificity, and Mn-/Fe-ABC uptake system usually do not discriminate very much between these to metal ions. That way, it could be excluded that the cells, TB-blocked and non-blocked, accumulate other metals due to the up-regulated import systems. For instance, although lactic acid bacteria usually do not use Fe, the Mn-depleted cells may accumulate Fe, which may inhibit LDH or induce synthesis of enzymes that interfere with lactic acid fermentation. An ICP-MS determination, rapidly done, should be very helpful to exclude such things.
3. Were controls of the experiments in Fig. 2 ff also done without erythromycin?
4. Minor: please insert an empty space between a number and its dimension according to ASM style.
5. LuxAB-assay. How was the presence of molecular oxygen managed?
6. Fig. 2. Deviations for the experiments in Panel B?
7. Fig. 2. How were "damaged cells" defined? I know them only to be alive or dead.
8. Fig. 4 is missing or Fig. 5 mislabeled.
9. Fig. 6. Maybe, adding other metals to the picture solves the riddle.
10. Table 1. Maybe, a cut-off of 2 for up- or down-regulation should be used.

Staff Comments:

Preparing Revision Guidelines

Please return the manuscript within 60 days; if you cannot complete the modification within this time period, please contact me. If you do not wish to modify the manuscript and prefer to submit it to another journal, please notify me of your decision immediately so that the manuscript may be formally withdrawn from consideration by Microbiology Spectrum.

Reviewer comments:

Reviewer #1 (Comments for the Author):

Manganese modulates metabolic activity and redox homeostasis in translationally-blocked *Lactococcus cremoris*, impacting metabolic persistence, cell-culturability, and flavor formation." (control no. Spectrum02708-21)

This work describes the impact of manganese limitation on the physiology of the model lactic acid bacterium *Lactococcus lactis*. Interestingly *L. lactis* can sustain strong growth in the absence of Mn. Proteomic analysis revealed Mn transporters to be upregulated upon Mn starvation, as well as several intracellular proteins involved in redox reactions. Using cells that are translationally blocked, it was found Mn omission leads to initially slower but more sustained lactic acid production and a greater conversion of aldehydes to alcohols during extended incubation. It was shown that the presence of Mn leads to NADH depletion which coincides with a significant loss of culturability.

Overall this manuscript provides new insights into the role Mn plays in cell physiology which significantly impacts *L. lactis* acidification, flavour compound production and survival. The model system established likely mimics the *L. lactis* non-growing status in a cheese ripening situation, and therefore is of interest to food fermentation applications and holds potential for future modulation of flavour compounds (e.g. through Mn addition to milk used in cheese making). The manuscript is very well written and data is clearly presented, and I don't have any major concerns.

AU (Authors): We would like to thank the reviewer for the compliments on our manuscript.

Specific comments:

- Line 255 and 264 - re-order the Figure 2C and 2D panels so that they are referred to in the text in the correct order.

AU: We have adjusted the figure panels and the text so that they are in the correct order.

- Some references are missing volume and page numbers (e.g. refs 9, 10, 40, 43, 44)

AU: We went through the reference list and corrected citations, so they comply with the required format.

Reviewer #2 (Comments for the Author):

The authors describe the impact of the presence of manganese in the growth medium, or the omission of Mn, on growth, acidification and survival of *Lactococcus cremoris*. A proteomic experiment shows that ABC- and NRAMP-type uptake systems are strongly up-regulated in the absence of Mn, as would have been expected. Since lactate dehydrogenase (LDH) should be activated by Mn(II) (or Co(II)) as in *Streptococcus faecalis*, absence of Mn should lead to missing activation of LDH, decreased acidification of the growth medium and accumulation of NADH. Surprisingly, this was not observed. The cells grew not differently in the presence or absence of Mn within 25 h, and the concentration of produced lactate was similar. Homofermentative lactate fermentation was done, nearly no acetate or formate produced by alternative fermentation pathways. The paper shifts subsequently to "translation-blocked" cells that were inhibited by erythromycin. Now, the cells produced only low amounts of lactate in the presence of Mn, did not show high numbers of dead cells in a life/dead straining despite a 100-fold decrease in cfu, indicating a vbcn state. These cells had only low NADH concentrations after 20 h. In contrast, they continued to produce lactate over days in the absence of Mn, possessed sufficient NADH after 20 h and contained a large portion of dead cells accompanied by a decrease of cfu after many days.

1. The Methods section is very small, in the paper and the supplement. The exact composition of the growth medium should appear in the supplement. The Methods in the paper should define "TB-cells" already here; it is defined too late in the Results part.

AU: We realize that the method section is very small in the main manuscript. However, we provided the relevant methods in the supplementary materials which span 3 pages and consist of over 1000 words. In the revised version, we included a table of the medium composition, as part of the Supplementary Methods section and adjusted the corresponding text in the paper.

As suggested, we also defined "TB-cells" already in line 94 of the Introduction section. As this definition might be overlooked, we additionally defined TB-cells in line 116 of the Methods section as well.

2. It could be very helpful to use an ICP-MS to measure the metal content of the growth medium "with omitted Mn" and also in the cells! NRAMP systems are of a broad substrate specificity, and Mn-/Fe-ABC uptake system usually do not discriminate very much between these two metal ions. That way, it could be excluded that the cells, TB-blocked and non-blocked, accumulate other metals due to the up-regulated import systems. For instance, although lactic acid bacteria usually do not use Fe, the Mn-depleted cells may accumulate Fe, which may inhibit LDH or induce synthesis of enzymes that interfere with lactic acid fermentation. An ICP-MS determination, rapidly done, should be very helpful to exclude such things.

AU: We thank the reviewer for this valuable feedback. We followed this suggestion and carried out additional experiments to quantify the levels of manganese and other transition metals (Fe, Co, and Zn) from the lysate and fresh media using ICP-MS. While we managed to obtain accurate measurements from the fresh media, the concentrations measured from lysates fell below the detection limit for all metals. This seems to indicate an issue with the digestion protocol of organic samples or matrix effects of the lysate. Consequently, we are not able to provide more information on the other metals that can be accumulated intracellularly in the absence of manganese.

3. Were controls of the experiments in Fig. 2 ff also done without erythromycin?

AU: The controls of the experiments in Figure 2 were only done with erythromycin. The reason for this is that prolonged incubation and measurement will not be feasible without erythromycin since growth would

lead to high cell numbers and pH inhibition within a few hours. This information can be found in Results subsection 3, line number 223 -227 as well.

4. Minor: please insert an empty space between a number and its dimension according to ASM style.

AU: We went through the text and adjusted these issues accordingly.

5. LuxAB-assay. How was the presence of molecular oxygen managed?

AU: Oxygen is supplied by allowing exposure to ambient air, which is sufficient for the luciferase reaction when a regular micro-plate lid is used instead of an optical sticker. This information has been added to the corresponding method section.

6. Fig. 2. Deviations for the experiments in Panel B?

AU: Figure 2 Panel B is composed of data from the replicates (n=3) which overlap in the figure. The variation between samples is therefore not visible in the plot. We updated the caption of the figure to clarify this point.

7. Fig. 2. How were "damaged cells" defined? I know them only to be alive or dead.

AU: We added information about the identification of damaged cells to the supplementary methods section:

"Live cells were characterized as population with high signal in FITC detector, but low signal in PE-Texas Red detector. Dead cells were characterized as the population with low signal in FITC detector, but high signal in PE-Texas Red detector. The third population of cells that was recognized displays high signal in both detectors, thereby placing these cells intermediately between the "live" and "dead" populations. Although the viability status and physiology of these cells is not entirely clear, we classified this population as "damaged cells" based on their staining characteristics."

8. Fig. 4 is missing or Fig. 5 mislabeled.

AU: We corrected the label and the text accordingly.

9. Fig. 6. Maybe, adding other metals to the picture solves the riddle.

AU: We agree that other metals may play a role and solve the riddle. Unfortunately, our ICP-MS results did not provide us with the relevant insights.

10. Table 1. Maybe, a cut-off of 2 for up- or down-regulation should be used.

AU: We thank the reviewer for the suggestion. We used the standard cutoff parameters in proteome analysis ($\alpha = 0.01$ and a False Discovery Rate (FDR) of 0.05). These cutoff parameters combine both fold change and the corrected p-value information. In our particular case introducing an additional cutoff for at least 2-fold change would result in the list being shortened by 5 proteins – of which 4 are relatively close to the proposed cutoff. As it is easy for the reader to identify those proteins in the table we chose to leave them in and maintain the standard proteome analysis methodology.

April 26, 2022

Dr. Herwig Bachmann
NIZO Food Research, TIFN
Kernhemseweg 2
Ede 6718ZB
Netherlands

Re: Spectrum02708-21R1 (Manganese modulates metabolic activity and redox homeostasis in translationally-blocked *Lactococcus cremoris*, impacting metabolic persistence, cell-culturability, and flavor formation.)

Dear Dr. Herwig Bachmann:

I am glad to inform you that your manuscript has been accepted, and I am forwarding it to the ASM Journals Department for publication. You will be notified when your proofs are ready to be viewed.

Sincerely,

Eva Sonnenschein
Editor, Microbiology Spectrum

Journals Department
Text S1: Accept
Table S1: Accept